# The TRIM25 Gene in Ducks: Cloning, Characterization and Antiviral Immune Response

**DOI:** 10.3390/genes13112090

**Published:** 2022-11-10

**Authors:** Jinlu Liu, Tiantian Gu, Jianzhou Chen, Shuwen Luo, Xiaoqian Dong, Ming Zheng, Guohong Chen, Qi Xu

**Affiliations:** 1Key Laboratory for Evaluation and Utilization of Poultry Genetic Resources of Ministry of Agriculture and Rural Affairs, Yangzhou University, Yangzhou 225009, China; 2Institute of Animal Husbandry and Veterinary Medicine, Zhejiang Academy of Agricultural Sciences, Hangzhou 310021, China

**Keywords:** duck, TRIM25, IFN-β, innate immune

## Abstract

**Simple Summary:**

TRIM25, as an important member of the TRIM family, has been frequently demonstrated in regulating the host’s antiviral response by activating innate immunity. Ducks are a natural reservoir and often serve as asymptomatic carriers of influenza A viruses, but the beneficial roles of TRIM25 in modulating the immune response remain largely unknown in ducks. On this basis, this study characterized *duTRIM25* and explored the role of different domains of duTRIM25 in duck antiviral immune response. The results displayed that the overexpression of duTRIM25 resulted in the IFN-β production after 5′ppp dsRNA infection and the SPRY domain of duTRIM25 partially enhanced duTRIM25’s antiviral activity. These data might reveal the functions of duTRIM25 in antiviral infections, providing new insights for understanding the protective immune responses of ducks.

**Abstract:**

TRIM25, as a significant member of the TRIM family, has been frequently demonstrated in regulating the host’s antiviral response by activating innate immunity. Ducks are often asymptomatic carriers of influenza A viruses, but the beneficial roles of TRIM25 in modulating the immune response remain largely unknown in ducks. In this study, we characterized the TRIM25, which contains a 16 bp 5′-UTR, a 279 bp 3′-UTR and a 2052 bp ORF that encodes 683 amino acid residues. In addition, we found that duTRIM25 transcripts were widely expressed in the 10 tissues tested, with higher expression levels in the kidney, liver, muscle and spleen and lower expression levels in the duodenum and blood. In addition, the six kinds of virus- or bacteria-mimicking stimuli were transfected into DEFs, and duTRIM25 was induced significantly with 5′ppp dsRNA stimulation. Furthermore, overexpression of duTRIM25 followed by treatment with 5′ppp dsRNA resulted in an increase in IFN-β. The SPRY domain of duTRIM25 contributed to promoting IFN-β activity in DEFs challenged with 5′ppp dsRNA. Taken together, our findings suggest that duck TRIM25 can induce the production of IFN-β against double-stranded RNA virus stimuli and that the SPRY domain of duTRIM25 was critical for the infection.

## 1. Introduction

More than 70 distinct protein members collectively form the tripartite motif (TRIM) protein family. Previous studies have demonstrated that many members of the TRIM family have antiviral effects. As early as 2004, scientists discovered that TRIM5α in African green monkeys and rhesus monkeys restricts the replication of not only the human immunodeficiency virus type 1, but also the N-tropic murine leukemia virus [1,2]. TRIM19 had also been shown to potentially interfere with the replication cycle of viruses. When TRIM19 was overexpressed, it exhibited the effect of resisting the influenza A virus and vesicular stomatitis virus [3]. Chicken TRIM25 was also found to inhibit the A avian leukosis virus replication by upregulating MDA5 receptor-mediated IFN-I response in chickens [4]. Collectively, the TRIM family is extensively involved in interferon production and viral replication by activating innate immunity.

The proteins of the TRIM family have a highly conserved N-terminal domain, including a RING domain, one or two B-box domains and a coiled-coil domain, which is often referred to as the RBBC [5,6]. On the other hand, they have a variable C-terminus binding to target proteins. PRY-SPRY motifs are the most common C-terminal TRIM domain and were first identified in TRIM27. However, these domains play different roles in different tissues or cells. Many studies have confirmed that RING domain-containing proteins can target specific proteins for degradation to activate specific signaling pathways, thereby depending on E3 ubiquitin ligase activity to regulate antiviral immune responses. Kallijarvi et al. showed that when the two conserved cysteines of the TRIM37 RING domain were converted to cysteines, the polyubiquitination was significantly reduced [7]. In addition, the RING domain of TRIM22 can inhibit long terminal repeat-driven transcription of human immunodeficiency virus 1 [8]. It has been reported that the B-box domains have zinc-binding motifs and also function as E3 ligases. Crucially, the B-box domains are unique to TRIM proteins and may be major determinants of this family [9]. TRIM15 lacking the B-box domain lost all antiviral activity. Actually, the B-box domain alone exhibited strong antiviral activity [10]. As all TRIM proteins have an RBCC motif, the capacity of the C-terminal domain to recruit specific partners plays an important role in their respective unique functions [11]. The SPRY domain was reported to participate in binding to RNA and interacting with proteins as well as interfering with viral replication. Li et al. have demonstrated that the loss of antiviral activity of TRIM62 was related to the deletion of the SPRY domain [12]. Therefore, the different members of the TRIMs have similar domains, but they rely on different domains to exert antiviral effects.

Ducks are a natural reservoir and often asymptomatic carriers of influenza A viruses. Previous studies showed that the immune system of ducks had an advantage in fighting weakly pathogenic avian influenza viruses. However, highly pathogenic influenza viruses appear to be able to escape attack by ducks’ immune systems [13]. Long-term co-evolution of ducks and influenza viruses fine-tunes many interactions between immune hosts and pathogens to confer resistance to disease. TRIM25, as an important member of the TRIM family, has been frequently demonstrated in regulating the host’s antiviral response by activating innate immunity. Wei et al. showed that goose TRIM25 transcription could be induced by goose IFN-α, IFN-γ, IFN-λ, poly(I:C), ODN 2006 and R848 in vitro, which means goose TRIM25 may play an active role in regulating antiviral immune responses [14]. Han et al. found that duck TRIM25 was upregulated in cells when treated with the duck Tembusu virus and inhibited viral expression [15]. These studies shed some light on duck TRIM25, but the beneficial roles of TRIM25 in modulating the immune response are still worth further exploration in ducks. In this study, we identified and characterized the duTRIM25 gene, and investigated the spatiotemporal expression profile of *duTRIM25*. Moreover, the six kinds of virus- or bacteria-mimicking stimuli were selected, and then the duTRIM25 transcripts were detected in DEFs challenged with the stimulus. Finally, the key domain of duTRIM25 was determined to promote IFN-β production. These data might reveal the functions of duTRIM25 in antiviral infections, providing new insights for understanding the protective immune responses of ducks.

## 2. Materials and Methods

### 2.1. Sample Collection and RNA Extraction

Five three-day-old, Shaoxing egg-laying ducks, having similar weight, were euthanized after anesthesia. The heart, liver, spleen, lung, kidney, duodenum, ovary, pancreas, muscle and blood of healthy five ducks were collected as samples. The samples were stored at minus 80 degrees, but blood was stored at minus 20 degrees for long-term storage until the next use.

According to the instructions of TRNzol Universal Reagent (TIANGEN, Beijing, China), the total RNA was extracted from different tissues of Shaoxing egg-laying ducks. The FastKing gDNA Dispelling RT SuperMix (TIANGEN) was used for reverse transcription to generate cDNA for RT-qPCR experiments.

### 2.2. DuTRIM25 cDNA Cloning and Sequence Analysis

Based on the duck *TRIM25* predicted sequences (GenBank accession number XM_005012325), the duTRIM25-FLAG-F and duTRIM25-FLAG-R primers were designed (Table 1), and the duTRIM25 cDNA was amplified from the spleen of duck by RT-PCR. Then the GeneRacer kit (L1500-01, Invitrogen, Carlsbad, CA) was used for obtaining the complete cDNA sequence of du*TRIM25* by 5′ and 3′ RACE.

A number of pieces of software were used to perform sequence analysis of TRIM25. Through the NCBI database, the sequences of TRIM25 in different species were found. A multiple sequence alignment of TRIM25 from representative species was performed using Clustal W. MEGA 6 software was used to perform phylogenetic analysis of the amino acid sequences of TRIM25 from 17 species. The conserved domains of duTRIM25 were predicted by NCBI Conserved Domains Database Tools.

### 2.3. Cell Culture and Virus Strain

The duck embryo fibroblasts (DEFs) were grown in complete Eagle’s Minimum Essential Medium (ATCC, Virginia, VA) supplemented with 10% fetal bovine serum (FBS) (Gibco, Grand Island, NY) [16]. The mimicking stimulations used in this study included LPS (InvivoGen, San Diego, CA), 5′ppp dsRNA (InvivoGen), Pam3CSK4 (InvivoGen), FLS-1 (InvivoGen), FLA-ST (InvivoGen) and R848 (Sigma, St Louis, MO). Six-well plates were used to seed DEFs until 70–80% confluency. Then DEFs were treated with the above mimicking stimulations. Cells were harvested after 24 h treatment and analyzed by RT-qPCR.

To further analyze the effect of 5′ppp dsRNA on the dynamic expression of duTRIM25, DEFs were seeded in six-well plates and raised to 70–80% confluency. The cells were transfected with 5′ppp dsRNA using Lipofectamine 2000 (Invitrogen, Carlsbad, CA) as specified by the manufacturer and then collected after 0, 3, 6, 9, 12 and 24 h of treatment. RT-qPCR was used to calculate duTRIM25 mRNA expression at different time points.

### 2.4. Plasmid Construction

According to the predicted IFN-β sequence of Anas platyrhynchos (GenBank accession no. KM032183), part of the sequence of the duck IFN-β promoter region (−167 to + 63) was selected to construct duIFN-β promoter. The duIFN-β promoter was cloned into the *Kpn*I and *Bgl*II sites of pGL3-Basic to form pGL3-IFN-β.

The ClonExpress II One Step Cloning Kit (Vazyme, Nanjing, China) was used to construct plasmids pcDNA3.1^+^-FLAG-duTRIM25, pcDNA3.1^+^-FLAG-duTRIM25(RING), pcDNA3.1^+^-FLAG-duTRIM25(B-Box) and pcDNA3.1^+^-FLAG-duTRIM25(SPRY). The constructed plasmids were handed over to TSINGKE Biological company for DNA sequencing verification. Table 1 lists the relevant primers.

### 2.5. Luciferase Reporter Assay

Firstly, 24-well plates were used to seed DEFs. When cells reached 50–70% confluence, the cells were transfected with 800 ng/well of plasmids, including pcDNA3.1^+^-FLAG-duTRIM25, pcDNA3.1^+^-FLAG-duTRIM25(RING), pcDNA3.1^+^-FLAG-duTRIM25(B-Box), pcDNA3.1^+^-FLAG-duTRIM25(SPRY) and 1 µg/well of reporter plasmid (pGL3-IFN-β); 50 ng/well of pRL-TK were co-transfected as an internal control vector. After 2 h, the transfection mixture was replaced with 500 μL of EMEM. Twenty-four hours post-transfection, the cells were treated with LPS (400 ng/well), 5′ppp dsRNA (500 ng/well) or normal saline as a control, and luciferase levels were measured 12 h later using Glomax (Promega, Madison, USA) [17].

### 2.6. Real-Time Quantitative PCR (RT-qPCR)

Primers required for RT-qPCR were designed by oligo 7 and then handed over to TSINGKE Biological Company for synthesis (Table 1). RT-qPCR experiments were used to detect changes in the mRNA levels of related genes. In the quantitative PCR system used in this experiment, the total volume of each sample was 10 μL, which contained 5 μL PowerUp SYBR Green Master Mix (Applied Biosystems, Thermo Fisher Scientific), 0.4 μL forward or reverse primer, 2 μL diluted cDNA template and 2.2 μL RNA-free water. The quantitative PCR program for this experiment was as follows: a holding step at 95 °C for 30 s, followed by 40 cycles of 10 s at 95 °C, and 30 s at 60 °C [18]. Duck β-actin was used as an internal reference, and the gene expression of the samples was analyzed using the 2^−ΔΔCt^ method.

### 2.7. Western Blotting Analysis

Five hundred microliters of a mixed solution consisting of cell lysis buffer (Biosharp, Guangzhou, China) for Western blotting and protease inhibitor (Beyotime, Shanghai, China) was added to lyse cells. In order to collect cellular proteins in the supernatant, the cell pellet was removed by centrifugation at 12,000 rpm at 4 °C for 10 min. After that, the samples were separated in 10% SDS-PAGE gel. Then, the extracted proteins were transferred to PVDF membranes, and 5% skim milk was used to block them. After that, the membranes were incubated with primary antibodies overnight at 4 °C and further incubated with an HRP-conjugated secondary antibody.

### 2.8. Statistical Analysis

Data are presented as mean ± standard error. *p* < 0.05 was considered statistically significant. GraphPad Prism 8.0 was used for processing figures.

## 3. Results

### 3.1. Sequence Analysis of Duck TRIM25

The full-length sequence of duck *TRIM25* cDNA was obtained using RT-PCR and RACE technology; the measured length was 2347 bp, which included a 2052 bp open reading frame (ORF) encoding 683 amino acids, a 16 bp 5′ UTR and a 279 bp 3′ UTR (KY_974316), The conserved domains of duck TRIM25 were analyzed using SMART, and we found that duTRIM25 as a protein of the TRIM family has a typical structure of RBBC, including the RING domain (20-63aa), two B-box domains (114-160aa, 163-194aa) and SPRY domain (511-679aa) (Figure 1A).

Based on the amino acid sequences of bird, mammalian and fish TRIM25, a phylogenetic tree was constructed in order to evaluate the molecular evolutionary relationship between duTRIM25 orthologs in other vertebrates (Figure 1B). DuTRIM25 shared high protein sequence similarity with *Anser cygnoides* TRIM25 (83.91%) and *Gallus gallus* TRIM25 (70.28%) but had a distant evolutionary relationship with mammalian TRIM25 and fish TRIM25. Consistent with the sequence alignment results, TRIM25 in birds is clustered in one group, and duTRIM25 was most closely related to the goose gene.

### 3.2. The Response to Different Stimulations on Duck TRIM25 Transcription Levels

To explore the regulation of *duTRIM25* expression in DEFs by different kinds of virus- or bacteria-mimicking stimuli, the *duTRIM25* mRNA levels were measured by RT-qPCR (Figure 2). The virus-mimicking stimuli are 5′ppp dsRNA and R848; the bacteria-mimicking stimuli are LPS, Pam3CSK4, FLS-1 and FLA-ST. The *TRIM25* mRNA levels were the highest after 5′ppp dsRNA treatment, followed by pam3CSK4 stimulations with higher expression.

### 3.3. The Spatiotemporal Expression Profile of TRIM25 in Ducks

In order to explore the spatial expression pattern of *duTRIM25* in different tissues, total RNA was extracted from various tissues of healthy Shaoxing egg-laying ducks, and the levels of duTRIM25 transcripts were detected by RT-qPCR. As shown in Figure 3a, *duTRIM25* is widely expressed in all tissues tested. Among them, *duTRIM25* is prominently expressed in the kidney, liver and muscle; moderately expressed in the spleen, heart, pancreas, lung and ovary; and expressed at relatively low levels in the duodenum and blood (Figure 3a).

To further determine the temporal changes in the expression of *duTRIM25* after 5′ppp dsRNA stimulation, DEFs were collected after 0, 3, 6, 9, 12 and 24 h of treatment with 5′ppp dsRNA and then detected by RT-qPCR. It turned out that the *duTRIM25* expression levels were significantly increased in DEFs within 6–24 h after 5′ppp dsRNA stimulation. The peak expression was reached at 12 h (Figure 3b). The results showed that the 5′ppp dsRNA infection affected *duTRIM25* mRNA expression. Specifically, *duTRIM25* mRNA levels were first upregulated and then downregulated, indicating that duTRIM25 negatively regulated the 5′ppp dsRNA replication.

### 3.4. Functional Analysis of the Induction of IFN-β by duTRIM25 and Its Different Domains

According to previous research results, we focused on the domains of duTRIM25, which could induce the expression of proinflammatory genes and further enhance the inflammatory response. The ORF of duTRIM25 has three different conserved domains, namely the RING domain, B-box domain and SPRY domain. Different domain segments are shown in Figure 4a. To explore the functions of duTRIM25 and its different domains, we cloned different duTRIM25 domains and constructed the expression plasmids TRIM25(RING), TRIM25(B-Box) and TRIM25(SPRY). Then, these recombinant plasmids were transfected into DEFs, and pcDNA3.1^+^-FLAG was transfected as a negative control. Western blot was used to detect the expression of different domains. The results showed that FLAG fusion proteins of different domains were all expressed in DEFs, as shown in Figure 4b.

To determine the effect of viral or bacterial irritation on IFN-β induction by duTRIM25, duTRIM25 was overexpressed in DEFs and then stimulated with 5′ppp dsRNA or LPS, respectively. The results showed that overexpression of duTRIM25 resulted in IFN-β production after 5′ppp dsRNA infection (Figure 4c). To further characterize the role of different duTRIM25 domains in the activation of the duck IFN-I signaling pathway, a dual-luciferase reporter assay was conducted to measure IFN-β promoter activity. The results indicated that TRIM25(SPRY) promoted IFN-β activity in DEFs challenged with 5′ppp dsRNA (Figure 4d). In general, SPRY domain overexpression partially enhanced duTRIM25′s antiviral activity; therefore, the SPRY domain may be an important structure of TRIM25 involved in the duck antiviral response.

## 4. Discussion

As an important member of the TRIM family, TRIM25 has been repeatedly reported to play important roles in mammalian immune responses to both viral and bacterial infections [19,20], which can induce the production of type I interferon and the upregulation of IFN-stimulated genes. In fact, *TRIM25* is also an important gene in duck antiviral immune response. The present findings support the hypothesis that TRIM25 can regulate the host’s antiviral response by activating innate immunity in ducks. In the study, we found that duck TRIM25 can induce the production of IFN-β against double-stranded RNA virus stimuli and that the SPRY domain of duTRIM25 was critical for the infection. These data provide references for understanding the mechanism of duTRIM25 exerting antiviral immune regulation.

We identified *duTRIM25* from Shaoxing egg-laying ducks. The full-length cDNA of duck *TRIM25* was cloned and sequenced; it contained a 2052 bp ORF, a 16 bp 5′-UTR and a 279 bp 3′-UTR. Duck TRIM25 conserved domains predicted by the CCD tool indicate that the duTRIM25 protein possesses typical features of TRIM family proteins, namely the RBCC domain, including one RING domain and two B-box-type zinc finger domains in the N-terminus and one SPRY domain in the C-terminus [21]. This is consistent with the results reported by Han et al. [15]. Previous studies have shown that TRIM22 relies on the RING domain to inhibit human immunodeficiency virus 1 long terminal repeat-driven transcription for antiviral effects [8]. However, the deletion of the B-box domain resulted in the loss of the antiviral activity of TRIM15 [10]. In addition, the antiviral activity of TRIM62 depends on the SPRY domain [12], which implies that the protein domain of the TRIM family may play a major role in recognizing pathogenic microorganisms and viruses and exerting antiviral immune responses.

The spatial expression profile of *TRIM25* in healthy duck tissues was evaluated in this study. *TRIM25* expression was monitored in all tissues tested, although expression levels differ across tissues (Figure 3a). The expression patterns of *TRIM25* in different tissues have been previously studied in poultry, including in chickens, ducks and geese. This study was consistent with the results showing that chicken, duck and goose *TRIM25* is widely expressed in a broad range of tissues and cells. Here, we summarized and analyzed the previous expression profiles of chickens, ducks and geese and the results of this study. Chicken *TRIM25* was highly expressed in the lung, thymus and spleen [22]. The highest goose *TRIM25* expression was detected in blood, proventriculus, Harderian gland and kidney [14]. Han et al. showed that duTRIM25 was highly expressed in the spleen, liver, trachea and muscle [15]. Because of the presence of IFN-stimulated response elements, TRIM25 expression was higher in immune-associated tissues. In the present study, we showed that *duTRIM25* was also expressed at higher levels in immune-related tissues such as the kidney, spleen and pancreas, suggesting that TRIM25 might play an important role in the innate immune response in ducks. Although the kidney is not an immune organ in ducks, it also has many immune functions. Sun et al. proved that when ducks were infected with a highly pathogenic avian influenza virus, the virus titer in the kidney of ducks was significantly increased [23]. Ou et al. showed that when duck kidney was attacked early by the avian hepatotropic virus, cytokine storms that included type I and type II IFNs were generated [24]. Interestingly, in addition to being highly expressed in immune-related tissues, *duTRIM25* was also dramatically expressed in the liver and muscle. TRIM59 has been confirmed to be highly expressed in the liver of mice, and it may be a potential therapeutic target for inflammatory diseases such as atherosclerosis [25]. Some studies suggest that TRIM25 is an E3 ligase that interacts with and degrades MTA-1 protein, which is one of the important mediators of metastatic progression in hepatocellular carcinoma [26]. Therefore, the high expression of duTRIM25 in the liver may imply that it also plays an important role in liver-related diseases. TRIM55 had been reported to act as a transient adapter between microtubules, actin and nascent myosin filaments and is involved in signaling from sarcomeres to the nucleus [27]. However, the role of TRIM25 in muscle has not been studied. We can only speculate that the high expression of duTRIM25 in muscle may be due to its potential regulatory effect on muscle sarcomere assembly.

The innate immune system is the host’s first line of defense against invading pathogens [28]. An essential part of the innate immune system is the host cells expressing pattern recognition receptors [29], which can detect pathogen-associated molecular patterns (PAMPs) and trigger the production of type I IFNs [30]. 5′ppp dsRNA is a virus mimic [31]; avian influenza virus infection can be simulated in vitro through it. It contains three phosphate groups on the 5’ end as its key element and thus can specifically activate the RLR signaling pathway to induce type I interferon production [32]. However, we must admit that there are still differences between the transfected virus mimics and avian influenza infection. Our research results provide a reference for the immune procedure in ducks after the simulation of avian influenza infection, but the findings in the actual infection situation have yet to be determined. In the next experiment, we will also use avian influenza infection for verification. Duck embryo fibroblasts are often used as a model for in vitro simulated virus infection experiments in ducks [33,34]; in this study, they also provided a preferred model for exploring the mechanisms of duTRIM25 function in avian innate immunity. However, DEFs also have limitations. In the present experiment, we found that the R848 treatment of DEFs failed to induce significant expression of duTRIM25. However, Wei et al. showed that the expression of goose TRIM25 was increased after R848 treatment in goose PBMCs. The reason for this may be that goose PBMCs are immune cells; compared to GEFs, they are more sensitive to immune response.

In the current study, we showed that 5′ppp dsRNA infection affected *duTRIM25* mRNA expression, first upregulating (at 12 h post-infection) and then downregulating (at 24 h post-infection) it in DEF cells, indicating that the host cells rapidly trigger antiviral defense to control infection within 12 h. Further, this shows the potential antiviral immune effect of duTRIM25 in RLR signaling pathways in duck cells.

As said before, the antiviral functions of TRIM proteins are closely related to their protein domains. To further explore the roles of different domains on IFN-β production, we detected the IFN-β dual-luciferase activity in DEFs transfected with the overexpression of different duTRIM25 domains. For full-length duTRIM25, a comparison of the luciferase levels driven after 5′ppp dsRNA treatment showed that the activity of IFN-β was increased significantly. Before and after treatment with 5′ppp dsRNA, the expression of the SPRY domain promoted IFN-β production. However, the luciferase levels of RING and B-box domains were always low. In summary, overexpression of the duTRIM25 SPRY domain in vitro can inhibit the replication of viruses by promoting the expression of IFN-β, illustrating that SPRY domain overexpression has the main impact on antiviral activity. As mentioned above, TRIM22 relies on the RING domain to inhibit HIV transcription [8]. However, the B-box domain is important for the antiviral activity of TRIM15 [10], while the deletion of the SPRY domain will make TRIM62 lose its antiviral activity [12]. From this, it can be concluded that though the TRIM family has the same conserved domains, different TRIM proteins rely on different domains to provide antiviral immunity. The SPRY domain is a prerequisite for the antiviral activity of duTRIM25. Further study is needed to explain the mechanism of the function of the duTRIM25 domains in the restriction of viral infection.

## 5. Conclusions

In conclusion, duTRIM25 transcripts were higher in the kidney, liver, muscle and spleen as well as lower in the duodenum and blood. DuTRIM25 was induced significantly with 5′ppp dsRNA stimulation compared with the other five kinds of virus- or bacteria-mimicking stimuli. Furthermore, overexpression of duTRIM25 followed by treatment with 5′ppp dsRNA resulted in increased IFN-β. The SPRY domain of duTRIM25 contributed to promoting IFN-β activity in DEFs challenged with 5′ppp dsRNA. These data might reveal the functions of duTRIM25 in antiviral infections, providing new insights into the protective immune responses of ducks.

## Figures and Tables

**Figure 1 genes-13-02090-f001:**
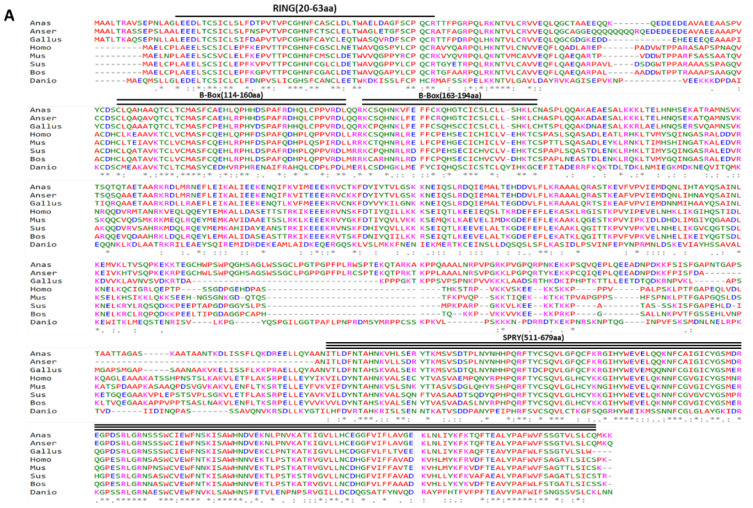
(**A**) Multiple alignments of TRIM25 protein sequences from duck (KY974316), goose (XP_047920213), chicken (NP_001305387.1), human (NP_005073.2), mouse (NP_033572.2), cattle (NP_001093806.1), pig (XP_005657028.3) and zebrafish (NP_956469.1). The symbol ‘*’ denotes 100% conserved residues, ‘:’ and ‘.’ represent relatively conservative residues, spaces represent non-conservative residues and ‘--’ indicates gaps during alignment. (**B**) A phylogenetic tree of the TRIM25 amino acid sequences was constructed by the neighbor-joining tree method using MEGA version 6. The sequence of each species was obtained from NCBI Genbank. The bar indicates the bootstrap value (%).

**Figure 2 genes-13-02090-f002:**
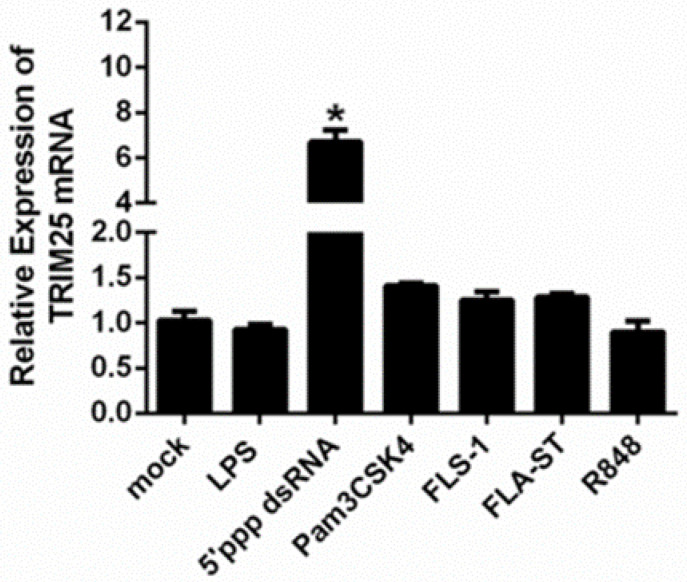
The expression of *duTRIM25* was analyzed in DEFs after different stimulations. LPS (lipopolysaccharides, the main components of the cell wall of Gram-negative bacteria), 5′ppp dsRNA (double-stranded RNA, which is a synthetic ligand for RIG-I), Pam3CSK4 (triacylated lipopeptide, which is the acylated amino terminus that mimics bacterial LPS), FLS-1 (Pam2CGDPKHPKSF, which is a synthetic lipoprotein derived from *Mycoplasma salivarium*), FLA-ST (flagellin from *Salmonella typhimurium*, the principal component of the flagella present on many bacteria), R848 (resiquimod, which is an imidazoquinoline compound with potent antiviral activity). Asterisks indicate significant differences determined by a *t*-test between mock and virus- or bacteria-mimicking stimulus treatments, *p* < 0.05. Each bar represents the mean ± SE (*n* = 3).

**Figure 3 genes-13-02090-f003:**
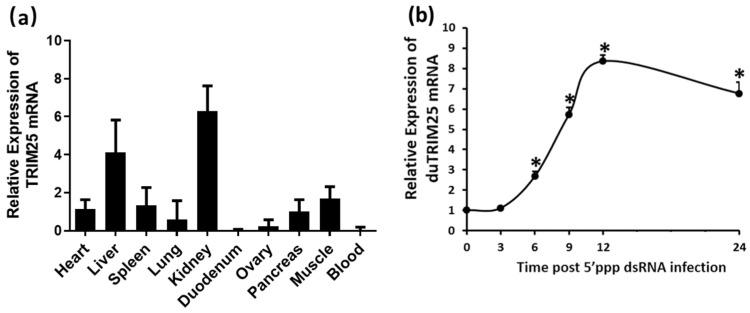
(**a**) Relative expression levels of *duTRIM25* in various tissues of five three-day-old healthy Shaoxing egg-laying ducks. The tissues tested included heart, liver, spleen, lung, kidney, duodenum, ovary, pancreas, muscle and blood. The mRNA expression levels were normalized to the expression of duck β-actin gene. Each bar represents the mean ± SE (*n* = 5). (**b**) Relative expression levels of *duTRIM25* in DEFs, treated with 5′ppp dsRNA and collected after 0, 3, 6, 9, 12 and 24 h. Each point represents the mean ± SE (*n* = 3). Asterisks indicate significant differences determined by a *t*-test between treatments 6, 9, 12 and 24 h and treatment 0 h (* *p* < 0.05).

**Figure 4 genes-13-02090-f004:**
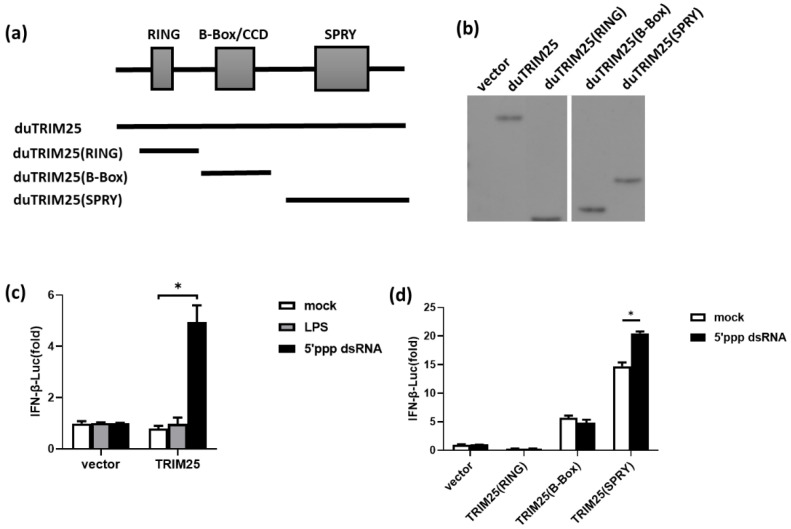
(**a**) Different duTRIM25 mutants were used to construct eukaryotic expression plasmids. TRIM25 represents the ORF of duTRIM25. TRIM25(RING) represents the RING domain of duTRIM25. TRIM25(B-Box) represents the B-box domain of duTRIM25. TRIM25(SPRY) represents the SPRY domain of duTRIM25. (**b**) DuTRIM25, duTRIM25(RING), duTRIM25(B-Box) and duTRIM25(SPRY) were all expressed in DEF cells. Expression of different mutants of duTRIM25 in DEF cells was measured by Western blot. DEF cells were transfected with the duTRIM25, duTRIM25(RING), duTRIM25(B-Box), duTRIM25(SPRY) and pcDNA3.1^+^-FLAG plasmids (1 ug each) for 24 h before Western blot analysis. Cell lysates were separated by SDS-PAGE, and different mutants of duTRIM25-FLAG were detected with mouse monoclonal antibodies against the FLAG tag. pcDNA3.1^+^-FLAG-transfected DEFs served as a negative control. (**c**) The induction of IFN-β by 5′ppp dsRNA or LPS was further determined. DEFs were co-transfected with pcDNA3.1^+^-FLAG-TRIM25 or vector together with pRL-TK *Renilla* luciferase plasmids and IFN-β luciferase reporter genes for 24 h, and stimulated with 5′ppp dsRNA, LPS or normal saline as a control for 12 h. Luciferase activity was then measured by a dual-luciferase reporter assay system. (**d**) Characterization of the effect of different domains of duTRIM25 on the induction of IFN-β. DEFs were co-transfected with expression plasmids TRIM25(RING), TRIM25(B-Box) and TRIM25(SPRY) together with pRL-TK *Renilla* luciferase plasmids and IFN-β luciferase reporter genes for 24 h and stimulated with 5′ppp dsRNA for 12 h. A dual-luciferase reporter assay system was used to measure luciferase activity. All data are presented as means ± SE (*n* = 3). Significance was analyzed by one-way ANOVA with Duncan method (* *p* < 0.05).

**Table 1 genes-13-02090-t001:** Primers used in the experiments.

Primer Name	Primer Sequence (5′→3′)	Annealing Temperature (°C)	Note
duTRIM25-FLAG-F	gggagacccaagctggctagcATGGCGGCGCTGACCCGA	68	RT-PCR
duTRIM25-FLAG-R	ctgctcggatcctttgaattcCTTTTTCATTTGGCAGAGGGAG		
5′ Router	AGGAGGCGCAGAAGTT	60	5′ RACE-PCR
5′ Inner	GTCACGGGGGTGTCGA		
3′ Router	TACCCAATGTGAGGGCTACCAAGAT	60	3′ RACE-PCR
3′ Inner	CTGCGAGGGAGGCTTTGTGATTTT		
duTRIM25(RING)-FLAG-F	gggagacccaagctggctagcATGGCGGCGCTGACCCGA	55	RT-PCR
duTRIM25(RING)-FLAG-R	ctgctcggatcctttgaattcCCGGCACAGCACCGTGTT		
duTRIM25(B-Box)-FLAG-F	gggagacccaagctggctagcATGAACAAGGTCTTCGAG	55	RT-PCR
duTRIM25(B-Box)-FLAG-R	ctgctcggatcctttgaattcAACAGTATAAATGTAATC		
duTRIM25(SPRY)-FLAG-F	gggagacccaagctggctagcATGTCAAAAGCTGCAACT	55	RT-PCR
duTRIM25(SPRY)-FLAG-R	ctgctcggatcctttgaattcCTTTTTCATTTGGCAGAGGGAG		
duIFN-β-promoter-F	CCCAAGCTTAAGCGATGGGAAAGATGT	58	RT-PCR
duIFN-β-promoter-R	GGAAGATCTTGTAGGGGCTATGTGGT		
qTRIM25-F	AAGCAGGAGGATGAGGAAGATGAGG	60	RT-qPCR
qTRIM25-R	AGAAGGAGGCCATGCAGGTCAG		
β-actin-F	ATGTCGCCCTGGATTTCG	60	RT-qPCR
β-actin-R	CACAGGACTCCATACCCAAGAA		

## Data Availability

Not applicable.

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
