# Peer review of "The TRIM25 Gene in Ducks: Cloning, Characterization and Antiviral Immune Response"

_genes, 2022, doi:10.3390/genes13112090_

Round 1
Reviewer 1 Report
In this manuscript, Liu et al. clone duck TRIM25 and sequence it. They determine the domains and the relative expression levels in different tissues. They also determine that overexpression increases stimulation of the interferon beta promoter (via luciferase) are increased upon stimulation with 5’ppp dsRNA. Changes in the manuscript for reporting of data are needed to make it easier to the reader. Additionally, the significance of the manuscript is decreased as there is no infection data to see if the interferon beta data means anything in terms of limiting viral replication (either when TRIM25 is overexpressed or removed).
Specific comments:
1. Why wasn’t infection performed to determine if duTRIM25 is a restriction factor/the interferon produced changes replication? This should be performed unless there is a reason that they cannot be done.
2. Figure 1-the different figures should be labeled A and B to correspond to text/figure legend.
3. Figure 3a-the lettering scheme is too hard for the reader to follow. Only show what is significant. Here, there isn’t a “standard” to compare to, so do statistics even matter?
4. Figure 4c-everything should be standardized to mock infected, to show the effect of stimulation. Redoing the math will show that the presence of something (like a virus) is showing an effect. If simply transfecting the SPRY domain stimulates the report (vs. needing LPS or dsRNA), that means nothing in the context of infection. It looks like only the full TRIM25 when standardized to mock might show any difference, but the data is needed.
5. Line 357: Remove “To sum up” (instead-In conclusion)
General comment:
The citations and references should all be the same format/be sure to check them (author, date in line 65 instead of number, reference 8 not containing last names of authors, etc.)
Author Response
Point 1: Why wasn’t infection performed to determine if duTRIM25 is a restriction factor/the interferon produced changes replication? This should be performed unless there is a reason that they cannot be done.
Response 1: Thank you for your suggestions. First of all, avian influenza is a zoonosis, and relevant experiments can only be conducted in biosafety level 3 laboratories. It is a pity that avian influenza cannot be used to infect in our laboratory. Secondly, the genome of avian influenza virus is composed of eight negative stranded RNA fragments and contains a 5 'triphosphate terminal. The RIG-I signal pathway can preferentially identify the RNA sequence with a 5' triphosphate terminal. Kowalinski et al. showed that 5' ppp dsRNA can specifically activate the RIG-I signaling pathway [1]. Therefore, in this study, 5'ppp dsRNA was used to simulate avian influenza virus infection and activate RIG-I signaling pathway. Thirdly, as can be seen from the results in Figure 4c, we had done experiments to determine that duTRIM25 is a restriction factor. We transfected the duTRIM25 plasmid into DEFs and treated with 5'ppp dsRNA or LPS, followed by dual-luciferase assays for IFN-β promoter activity. We found that overexpression of duTRIM25 promoted IFN-β promoter activity induced by 5’ppp dsRNA. It can be seen that duTRIM25 significantly upregulates type I interferon transcription induced by 5’ppp dsRNA.
Figure 4. (c) The induction of IFN-β by 5 'ppp dsRNA or LPS was further determined. DEFs were co-transfected with pcDNA3.1+-FLAG-TRIM25 or vector together with pRL-TK renilla luciferase plasmids, and IFN-β luciferase reporter genes for 24 h, and stimulated with 5’ppp dsRNA or LPS for 12 h. Luciferase activity was then measured by a dual-luciferase reporter assay system. All luciferase activity was normalized to renilla luciferase activity. All data are presented as means ± SE from at least three independent triplicate experiments. Significance was analyzed by one‐way ANOVA with Duncan method (*p < 0.05).
Reference
- Kowalinski, E.; Lunardi, T.; McCarthy, A.A.; Louber, J.; Brunel, J.; Grigorov, B.; Gerlier, D.; Cusack, S. Structural Basis for the Activation of Innate Immune Pattern-Recognition Receptor RIG-I by Viral RNA. Cell 2011, 147, 423-435.
Point 2: Figure 1-the different figures should be labeled A and B to correspond to text/figure legend.
Response 2: Thank you for your suggestions. We have labeled the corresponding figure A and B to make them clearer and easier to understand.
Point 3. Figure 3a-the lettering scheme is too hard for the reader to follow. Only show what is significant. Here, there isn’t a “standard” to compare to, so do statistics even matter?
Response 3: Thank you for your suggestions. No standard group can be compared in tissue expression profile. It showed the relative expression of duTRIM25 in different tissues. The lettering scheme can show the relationship between two tissues.
Figure 3. (a) Relative expression levels of duTRIM25 in various tissues of five three-day-old healthy Shaoxing egg-laying ducks. The tissues tested included duodenum, ovary, pancreas, muscle and blood. The mRNA expression levels were normalized to the expression of duck β-actin gene. Each bar represents the mean ± SE (n=5). Same letters meant no significant difference(p>0.05), different letters meant significant difference (p<0.05).
Point 4. Figure 4c-everything should be standardized to mock infected, to show the effect of stimulation. Redoing the math will show that the presence of something (like a virus) is showing an effect. If simply transfecting the SPRY domain stimulates the report (vs. needing LPS or dsRNA), that means nothing in the context of infection. It looks like only the full TRIM25 when standardized to mock might show any difference, but the data is needed.
Response 4: Thank you for your advice. We have reanalyzed and made the figure again. As you said, some effects did not show up until redo the math. Therefore, we also rewrite the results 3.4. ‘To determine the effect of viral or bacterial irritation on IFN-β induction by duTRIM25, duTRIM25 was overexpressed in DEFs and then stimulated with 5’ppp dsRNA or LPS, respectively. The results showed that overexpression of duTRIM25 resulted in the IFN-β production after 5’ppp dsRNA infection (Figure 4c). To further characterize the role of different duTRIM25 domains in the activation of duck IFN-I signaling pathway, a dual luciferase reporter assay was conducted to measure IFN-β promoter activity. The results indicated that TRIM25(SPRY) promoted IFN-β activity in DEFs challenged with 5’ppp dsRNA (Figure 4d). In general, overexpression SPRY domain partially enhanced duTRIM25’s antiviral activity, therefore SPRY domain may be important structure of TRIM25 involved in duck antiviral response.’ (Line 253-262).
Figure 4. (c) The induction of IFN-β by 5 'ppp dsRNA or LPS was further determined. DEFs were co-transfected with pcDNA3.1+-FLAG-TRIM25 or vector together with pRL-TK renilla luciferase plasmids, and IFN-β luciferase reporter genes for 24 h, and stimulated with 5’ppp dsRNA, LPS or normal saline as a control for 12 h. Luciferase activity was then measured by a dual-luciferase re-porter assay system. (d) Characterization of the effect of different domains of duTRIM25 on the induction of IFN-β. DEFs were co-transfected with expression plasmids of TRIM25(RING), TRIM25(B-Box), TRIM25(SPRY) together with pRL-TK renilla luciferase plasmids, and IFN-β luciferase reporter genes for 24 h, and stimulated with 5’ppp dsRNA for 12 h. A dual luciferase reporter assay system was used to measure luciferase activity. All data are presented as means ± SE (n=3). Significance was analyzed by one‐way ANOVA with Duncan method (*p < 0.05).
Point 5. Line 357: Remove “To sum up” (instead-In conclusion)
Response 5: Thank you for your suggestions. We have changed ‘To sum up’ to ‘In conclusion’. (Line 378).
Point 6. The citations and references should all be the same format/be sure to check them (author, date in line 65 instead of number, reference 8 not containing last names of authors, etc.)
Response 6: Thank you for your suggestions. We have checked every citation and reference carefully to make sure they were in the correct format for GENES.

Reviewer 2 Report
TRIM25, a E3 ubiquitin ligase, modulates the antiviral response by ubiquitinating cytosolic PRRs such as RIG-1. In this manuscript Liu et al., report the cloning and functional characterization of TRIM25 cDNA from Shaoxing egg-laying ducks. They also analyzed the TRIM25 mRNA expression in various duck tissues and in duck embryo fibroblasts under various antigenic stimuli like 5’ppp dsRNA, LPS, etc. Eukaryotic FLAG-tagged expression constructs of full duTRIM25 and partial domains like RING, B-BOX, SPRY, etc were also made; DEF transfected with these. The IFN-β activity was measured by Renilla luciferase assay.
A previous report reported on the identification of TRIM25 gene and expression of its tissue in Duck (Kaikai et al, 2021). So the gene cloning and tissue expression are not novel. However, the authors have investigated other aspects like the expression of TRIM25 in DEF in response to various stimuli like 5’ppp dsRNA, LPS, etc. They also mapped the functional activity of different domains of TRIM25 by transfecting FLAG-tagged expression constructs and analyzing IFN-b promoter activity. Methods used in this manuscript are standard for this kind of studies, the paper has been well-written, and data have been analyzed using appropriate statistical techniques. Below are a few suggestions for further improving the manuscript.
Major Comments
Major comment 1 : Previous reports of duck TRIM25 and Goose TRIM25 not cited or mentioned. Kindly mention, cite and compare: Kaikai et al, 2021 (https://doi.org/10.3389/fvets.2021.722113 ) have reported the identification of duck TRIM25 with the 2052 bp ORF encoding a 684 amino acid protein with RING finger domain, B-box domain, a coiled-coil domain, and PRY/SPRY domain. They also reported the gene expression of duck TRIM25 in various duck tissues, and in virus-infected and IFN-treated duck fibroblasts. So, these results have already been reported. Moreover, the authors have not cited this prior work in the introduction/ discussion. The authors are requested to compare the cloned sequence with the already reported duck TRIM25 and identity and report any novel variations of significance if any. Otherwise, the identification and sequencing of TRIM25 would be just a repeat of the earlier work (Kaikai et al, 2021).
Also, TRIM25 from Goose (Anser cygnoides), another member of the avian order Anseriformes has been reported (Wei et al, 2016) (https://doi.org/10.1155/2016/1403984). Please cite this paper in the introduction and compare the goose sequence with the duck sequence and indicate the similarity and highlight any unique changes in duck TRIM25. Kindly compare the expression pattern of goose TRIM25 with the current study and highlight any significant differences with possible reasons in the discussion. The sequence TRIM25 ortholog from goose also needs to be compared in the sequence alignment (Fig 1a) and phylogenetic tree (Fig 1b).
Both these papers (Kaikai et al, 2021) & (Wei et al, 2016) should be cited and compared with the current study.
Major comment 2: Figure 1a update with the other previous reported Goose and Duck sequence: It would be better if the authors can include the previous report duck sequence (Kaikai et al, 2021,) and the goose sequence (Wei et al, 2016), along with the authors TRIM25 from Shaoxing egg-laying ducks and highlight any significant amino acid sequence differences between these. In addition, the order of the species in the alignment should be changed to make more sense based on phylogenetic relationships (The avian TRIM25 sequences should be together in the multiple alignments, ie: duck, goose, hen, first followed by reptiles/mammals).
Major comment 3: Tissue expression- Figure 3a – The authors report a very high expression of TRIM25 in the kidney (~ 2000 fold). In previous reports of duck TRIM25 (Kaikai et al, 2021) and goose TRIM25 (Wei et al., 2016), the expression in the kidney is relatively low (around 2 fold). The authors are requested to comment on this anomalous 103-fold higher expression in their studies. In this study, the expression in the liver is also log-fold higher compared to earlier reports in other studies.
Also in the tissue expression study, blood was not analyzed. This is a significant omission considering the report that TRIM25 was most highly expressed in the goose blood in both young as well as adult geese (Wei et al, 2016). The previous report on duck TRIM25 (Kaikai et al, 2021) did not seem to assess the expression in blood. If the authors have data on duck TRIM25 blood expression, it would have provided novel information.
Major comment 4: Details of Tissue expression RT-qPCR lacking: Key information on RT-qPCR for the data shown in Figure 3a like primers used for real-time PCR, and the tissue sample used as a calibrator for the tissue expression experiments. Table 1 contains only RACE primers and FLAG-tagged primers (for eukaryotic expression). Authors are request to provide these.
Major comment 5: 5’ppp dsRNA and RIG-I, method of delivery: Chicken does not have the RIG-I gene, whereas duck has RIG-I. 5’ppp dsRNA is a key ligand (agonist) for the cytosolic sensor RIG-I. And the immune stimulation of RIG-I by 5’ppp dsRNA depends on the method of delivery into the cell. Here authors have not provided details on the methods ( Section 2.4) used to deliver 5’ppp dsRNA into DEF for the data shown in Figure 3b. Transfection and direct treatment with 5’ppp dsRNA or Poly (I:C) will have different outcomes as the methods of delivery dictate the receptor stimulated. If 5’ppp dsRNA was transfected, there should be a mock transfection control (just the transfection reagent) also to compare the effects of any non-specific immune stimulation due to the transfection.
In Goose cells, R848 treatment increased the TRIM25 expression significantly (Wei et al, 2016), whereas DEF treated with R848 did not produce significant TRIM25 expression (Figure 2). If the authors can discuss why the TLR agonist R848 failed to induce significant TRIM25 expression it would be nice.
Minor comments
Line 80-81- ‘the beneficial roles of TRIM25 in 80 modulating the immune response are poorly understood in duck…’ - The authors should recheck this statement in the introduction based on the prior report of duck TRIM25 (Kaikai et al, 2021).
Lines 103, 206, 268, - Please italicize all scientific names (binomial nomenclature) of organisms
Line 137 – Reference should be provided for the pGL-IFN-β reporter plasmid
Line 143-145 – ELISA is mentioned in the methods- Results or Discussion do not mention the results of the Duck NF-κB and IRF1 ELISA. Please provide details.
Line 263, 265, etc –‘Flag’ – Please change to ‘FLAG’ while referring to the FLAG-tag
Author Response
Point 1: Major comment 1: Previous reports of duck TRIM25 and Goose TRIM25 not cited or mentioned. Kindly mention, cite and compare: Kaikai et al, 2021 (https://doi.org/10.3389/fvets.2021.722113 ) have reported the identification of duck TRIM25 with the 2052 bp ORF encoding a 684 amino acid protein with RING finger domain, B-box domain, a coiled-coil domain, and PRY/SPRY domain. They also reported the gene expression of duck TRIM25 in various duck tissues, and in virus-infected and IFN-treated duck fibroblasts. So, these results have already been reported. Moreover, the authors have not cited this prior work in the introduction/ discussion. The authors are requested to compare the cloned sequence with the already reported duck TRIM25 and identity and report any novel variations of significance if any. Otherwise, the identification and sequencing of TRIM25 would be just a repeat of the earlier work (Kaikai et al, 2021).
Also, TRIM25 from Goose (Anser cygnoides), another member of the avian order Anseriformes has been reported (Wei et al, 2016) (https://doi.org/10.1155/2016/1403984). Please cite this paper in the introduction and compare the goose sequence with the duck sequence and indicate the similarity and highlight any unique changes in duck TRIM25. Kindly compare the expression pattern of goose TRIM25 with the current study and highlight any significant differences with possible reasons in the discussion. The sequence TRIM25 ortholog from goose also needs to be compared in the sequence alignment (Fig 1a) and phylogenetic tree (Fig 1b).
Both these papers (Kaikai et al, 2021) & (Wei et al, 2016) should be cited and compared with the current study.
Response 1: Thank you for your suggestions.
① First of all, I must admit that I was negligent in not paying attention to this article(Kaikai et al, 2021). At the beginning of the relevant work, we did not retrieve the article. We have completed all the experiments before 2020. Under your kind reminder, I have carefully read and studied this article and cited this prior work in the introduction and discussion. In the introduction section, we have made the following additions. ‘Kaikai et al found that duck TRIM25 was upregulated in cells when treated by duck Tembusu virus and inhibited viral expression.’ (Line 82-83). In the discussion section, we also show agreement with the previous findings of KaiKai et al. ‘We identified duTRIM25 from Shaoxing egg-laying ducks. The full-length cDNA of duck TRIM25 had been cloned and sequenced, containing a 2052-bp ORF, a 16-bp 5′-UTR and a 279-bp 3′-UTR. Duck TRIM25 conserved domains predicted by the CCD tool indicate that the duTRIM25 protein possessed typical features of TRIM family proteins - RBCC domain, including one RING domain and two B-Box-type zinc finger domains in N-terminal, one SPRY domain in C-terminal [20]. This is consistent with the results reported by Kai-Kai et al.’ (Line 296-302)
② I must admit that there are some overlapping parts in our work. Kaikai et al. have reported that the identification of duck TRIM25 with the 2052 bp ORF encoding a 684 amino acid protein, but we further determined that duck TRIM25 has a 16-bp 5′ UTR, a 279-bp 3′ UTR by RACE technology. Our study provided further insight into the full-length sequence of the duck TRIM25 sequence and is not merely a repeat of earlier work.
③ The reviewer suggested that we should compare the cloned sequence with the Kaikai et al already reported duck TRIM25 and report any novel variations of significance. We think it's a brilliant idea, hopefully to compare the subtle differences in TRIM25 between different duck breeds. Unfortunately, Kaikai et al only mentioned the cloned sequence of duck TRIM25 in the article, but did not list the complete sequence, so no comparison could be made.
④ Both geese and ducks are waterfowl. The study of goose TRIM25 was very valuable for this article. Thank you for reminding me that I have cited the paper (Wei et al, 2016) in the introduction. Wei et al. showed that goose TRIM25 transcription could be induced by goose IFN-α, IFN-γ, IFN-λ, poly(I:C), ODN 2006, and R848 in vitro, it means goose TRIM25 may play an active role in regulating antiviral immune responses. (Line79-82). I have compared the expression pattern of goose TRIM25 with the present study in the discussion. (Line 317-318). We have included the goose TRIM25 sequence in the sequence alignment (Figure 1A) and phylogenetic tree (Figure 1B) for comparison.
Point 2: Figure 1a update with the other previous reported Goose and Duck sequence: It would be better if the authors can include the previous report duck sequence (Kaikai et al, 2021,) and the goose sequence (Wei et al, 2016), along with the authors TRIM25 from Shaoxing egg-laying ducks and highlight any significant amino acid sequence differences between these. In addition, the order of the species in the alignment should be changed to make more sense based on phylogenetic relationships (The avian TRIM25 sequences should be together in the multiple alignments, ie: duck, goose, hen, first followed by reptiles/mammals).
Response 2: Thank you for your suggestions.
① We have added the previous report goose TRIM25 sequence (Wei et al, 2016) and Shaoxing egg-laying ducks TRIM25 sequence into the amino acid sequence alignment. Amino acid differences of TRIM25 in different species were demonstrated by different symbols. Compared with the amino acid sequences of TRIM25 in other species, the sequence of duTRIM25 presented a much higher identity with that of goose (83.91%) and chicken (70.28%)
② We have reordered the species in the phylogenetic tree according to their phylogenetic relationships. Because the sequences of goose TRIM25 were added to construct the phylogenetic tree, the phylogenetic relationships were also slightly adjusted. Thus, we corrected the sentence’ Based on the amino acid sequences of mammalian, bird, and fish TRIM25, a phylogenetic tree was constructed in order to evaluate the molecular evolutionary relationship between duTRIM25 orthologues in other vertebrates (Figure 1B). DuTRIM25 shared high protein sequence similarity with Anser cygnoides TRIM25 (83.91%) and Gallus gallus TRIM25 (70.28%), but had a distant evolutionary relationship with mammalian TRIM25 and fish TRIM25. Consistent with the sequence alignment results, TRIM25 in birds is clustered in one group, and duTRIM25 was most closely related to goose gene.’ (Line182-188).
Figure 1. (A) Multiple alignments of TRIM25 protein sequences from duck (KY974316), goose (XP_047920213), chicken (NP_001305387.1), human (NP_005073.2), mouse (NP_033572.2), cattle (NP_001093806.1), pig (XP_005657028.3), and zebrafish (NP_956469.1). The symbols ‘﹡’ denote 100% conserved residues, ‘:’ and ‘.’ represent relatively conservative residues, spaces represent non-conservative residues, ‘--’ indicates gaps during alignment. (B) Phylogenetic tree of the TRIM25 amino acid sequences. The tree was constructed by the neighbor-joining tree method using amino acid sequences aligned with MEGA version 6. The bar indicates the bootstrap value (%).
Point 3: Tissue expression- Figure 3a– The authors report a very high expression of TRIM25 in the kidney (~ 2000 fold). In previous reports of duck TRIM25 (Kaikai et al, 2021) and goose TRIM25 (Wei et al., 2016), the expression in the kidney is relatively low (around 2 fold). The authors are requested to comment on this anomalous 103-fold higher expression in their studies. In this study, the expression in the liver is also log-fold higher compared to earlier reports in other studies.
Also in the tissue expression study, blood was not analyzed. This is a significant omission considering the report that TRIM25 was most highly expressed in the goose blood in both young as well as adult geese (Wei et al, 2016). The previous report on duck TRIM25 (Kaikai et al, 2021) did not seem to assess the expression in blood. If the authors have data on duck TRIM25 blood expression, it would have provided novel information.
Response 3: Thank you for your suggestions.
①First of all, although the kidney is not an immune organ in ducks, it also plays a lot of immune functions. Hailiang et al. have proved that when ducks were infected with highly pathogenic avian influenza virus, the virus titer in the kidney of ducks was significantly increased[1]. Xumin et al. showed that when duck kidney was attacked early by the avian hepatotropic virus, cytokine storms were generated such as type I and type II IFNs[2]. TRIM25, as an important component of RIG-I signaling pathway and one of the interferon stimulated genes, it is also reasonable that high expression in immune-related tissues. Secondly, gene expression is species and breed specific. We found that immune-related genes were also highly expressed in the kidney in other articles that measured gene expression in various tissues of Shaoxing ducks[3].
②Tissue expression studies are relatively quantitative calculations. Because of the lack of a control group, we usually select an organization as control group. In this paper, we selected the duodenum with the lowest expression as the control, so as to calculate the expression of other tissues relative to the duodenum. The advantage of this is that the pictures are very beautiful, so the expression of the tissue is reflected by the height of the column. However, it can also lead to misunderstandings, such as what appears to be duTRIM25 were log-fold higher in the kidney and liver compared to earlier reports in other studies. In order to eliminate this misunderstanding, we chose the moderately expressed heart as the new control, and re-calculated and drew the figure3a. As you can see, the new figure3a is better understood.
Figure 3. (a) Relative expression levels of duTRIM25 in various tissues of five three-day-old healthy Shaoxing egg-laying ducks. The tissues tested included duodenum, ovary, pancreas, muscle and blood. The mRNA expression levels were normalized to the expression of duck β-actin gene. Each bar represents the mean ± SE (n=5). Same letters meant no significant difference(p>0.05), different letters meant significant difference (p<0.05).
③We have supplemented the expression of duTRIM25 in duck blood. The methods of blood preservation and RNA extraction were also supplemented in the materials and methods section. (Line 94-102)
References
- Sun, H.; Jiao, P.; Jia, B.; Xu, C.; Wei, L.; Shan, F.; Luo, K.; Xin, C.; Zhang, K.; Liao, M. Pathogenicity in quails and mice of H5N1 highly pathogenic avian influenza viruses isolated from ducks. Vet Microbiol 2011, 152, 258-265.
- Ou, X.; Mao, S.; Jiang, Y.; Zhang, S.; Ke, C.; Ma, G.; Cheng, A.; Wang, M.; Zhu, D.; Chen, S.; et al. Viral-host interaction in kidney reveals strategies to escape host immunity and persistently shed virus to the urine. Oncotarget 2017, 8, 7336-7349.
- Gu, T.; Li, G.; Wu, X.; Zeng, T.; Xu, Q.; Li, L.; Vladyslav, S.; Chen, G.; Lu, L. Molecular cloning, tissue distribution and function analysis of duck TLR7. Anim Biotechnol 2022, 33, 234-241.
Point 4: Details of Tissue expression RT-qPCR lacking: Key information on RT-qPCR for the data shown in Figure 3a like primers used for real-time PCR, and the tissue sample used as a calibrator for the tissue expression experiments. Table 1 contains only RACE primers and FLAG-tagged primers (for eukaryotic expression). Authors are request to provide these.
Response 4: Thank you for your suggestions. All primers used for RT-qPCR in this paper have been added to Table1.
Table 1. Primers used in the experiments.
|
Primer name |
Primer sequence (5’→3’) |
Annealing temperature(℃) |
Note |
|
duTRIM25-FLAG-F |
gggagacccaagctggctagcATGGCGGCGCTGACCCGA |
68 |
RT-PCR |
|
duTRIM25-FLAG-R |
ctgctcggatcctttgaattcCTTTTTCATTTGGCAGAGGGAG |
||
|
5’ Router |
AGGAGGCGCAGAAGTT |
60 |
5’ RACE-PCR |
|
5’ Inner |
GTCACGGGGGTGTCGA |
||
|
3’ Router |
TACCCAATGTGAGGGCTACCAAGAT |
60 |
3’ RACE-PCR |
|
3’ Inner |
CTGCGAGGGAGGCTTTGTGATTTT |
||
|
duTRIM25(RING)-FLAG-F |
gggagacccaagctggctagcATGGCGGCGCTGACCCGA |
55 |
RT-PCR |
|
duTRIM25(RING)-FLAG-R |
ctgctcggatcctttgaattcCCGGCACAGCACCGTGTT |
||
|
duTRIM25(B-Box)-FLAG-F |
gggagacccaagctggctagcATGAACAAGGTCTTCGAG |
55 |
RT-PCR |
|
duTRIM25(B-Box)- FLAG-R |
ctgctcggatcctttgaattcAACAGTATAAATGTAATC |
||
|
duTRIM25(SPRY)- FLAG-F |
gggagacccaagctggctagcATGTCAAAAGCTGCAACT |
55 |
RT-PCR |
|
duTRIM25(SPRY)-FLAG-R |
ctgctcggatcctttgaattcCTTTTTCATTTGGCAGAGGGAG |
||
|
duIFN‐β‐promoter‐F |
CCCAAGCTTAAGCGATGGGAAAGATGT |
58 |
RT‐PCR |
|
duIFN‐β‐ promoter‐R |
GGAAGATCTTGTAGGGGCTATGTGGT |
||
|
qTRIM25‐F |
AAGCAGGAGGATGAGGAAGATGAGG |
60 |
RT‐qPCR |
|
qTRIM25‐R |
AGAAGGAGGCCATGCAGGTCAG |
||
|
β-actin-F |
ATGTCGCCCTGGATTTCG |
60 |
RT‐qPCR |
|
β-actin-R |
CACAGGACTCCATACCCAAGAA |
Point 5: 5’ppp dsRNA and RIG-I, method of delivery: Chicken does not have the RIG-I gene, whereas duck has RIG-I. 5’ppp dsRNA is a key ligand (agonist) for the cytosolic sensor RIG-I. And the immune stimulation of RIG-I by 5’ppp dsRNA depends on the method of delivery into the cell. Here authors have not provided details on the methods (Section 2.4) used to deliver 5’ppp dsRNA into DEF for the data shown in Figure 3b.
Transfection and direct treatment with 5’ppp dsRNA or Poly (I:C) will have different outcomes as the methods of delivery dictate the receptor stimulated. If 5’ppp dsRNA was transfected, there should be a mock transfection control (just the transfection reagent) also to compare the effects of any non-specific immune stimulation due to the transfection.
In Goose cells, R848 treatment increased the TRIM25 expression significantly (Wei et al, 2016), whereas DEF treated with R848 did not produce significant TRIM25 expression (Figure 2). If the authors can discuss why the TLR agonist R848 failed to induce significant TRIM25 expression it would be nice.
Response 5: Thank you for your suggestions.
① We have added detailed methods for delivering 5’ppp dsRNA into DEFs about Figure 3b. The specific supplementary content is as follows ‘DEFs were seeded in six-well plates and raised to 70-80% confluency. The cells were transfected with 5'ppp dsRNA using Lipofectamine 2000 (Invitrogen, USA) as specified by the manufacturer and then collected after 0, 3, 6, 9, 12, 24h treatment. RT-qPCR was used to calculate duTRIM25 mRNA expression at different time points.’ (Line 126-130)
② As shown in Figure 2, We compared the DEFs transfected with 5'ppp dsRNA to mock transfection control. It can be seen that transfection of 5’ppp dsRNA can significantly increase the expression level of duTRIM25 compared with mock group. The next step was shown in Figure 3b, we further explored the dynamic effect of 5’ppp dsRNA transfection on duTRIM25 expression at different time points. We found that the expression of duTRIM25 increased first and then decreased, and the expression of duTRIM25 reached its peak at 12h after transfection with 5’ppp dsRNA. Unfortunately, in this experiment, we did not detect the expression of duTRIM25 only after transfection with transfection reagents at different time points to exclude the effect of any non-specific immune stimulation caused by transfection.
③ Wei et al showed that the expression of goose TRIM25 was increased after R848 treatment in goose PBMCs. We found that the authors also did not attempt to detect the ability of R848 to induce goose TRIM25 in GEFs. Goose PBMCs are immune cells. Compared to GEFs, it is more sensitive and rapid to immune response. Study found that R848 agonists are potent inducers of PBMC‑produced cytokines that inhibit hepatitis B virus production in primary human hepatocytes[4]. In the present experiment, we found that R848 treatment of DEFs failed to induce significant expression of duTRIM25. In the present experiment, we found that R848 treatment of DEFs failed to induce significant expression of duTRIM25. We believe that it is not that R848 cannot induce significant expression of duTRIM25, but only that it cannot induce significant expression of TRIM25 in DEFs. In the next experiments, we will try to perform related experiments in duck PBMCs to demonstrate a more comprehensive immune response. In the discussion, we also added the following statement. ‘But DEFs also has its limitations. In the present experiment, we found that R848 treatment of DEFs failed to induce significant expression of duTRIM25. However, Wei et al showed that the expression of goose TRIM25 was increased after R848 treatment in goose PBMCs. The reason may be goose PBMCs are immune cells, compared to GEFs, it is more sensitive and rapid to immune response.’ (Line 349-354)
References
- Janovec, V.; Hodek, J.; Clarova, K.; Hofman, T.; Dostalik, P.; Fronek, J.; Chlupac, J.; Chaperot, L.; Durand, S.; Baumert, T.F.; et al. Toll-like receptor dual-acting agonists are potent inducers of PBMC-produced cytokines that inhibit hepatitis B virus production in primary human hepatocytes. Sci Rep 2020, 10, 12767.
Point 6: Line 80-81- ‘the beneficial roles of TRIM25 in modulating the immune response are poorly understood in duck…’ - The authors should recheck this statement in the introduction based on the prior report of duck TRIM25 (Kaikai et al, 2021).
Response 6: Thank you for your suggestions. We have changed the statement in the introduction. ‘Kaikai et al found that duck TRIM25 was upregulated in cells when treated by duck Tembusu virus and inhibited viral expression. This study shed some light on duck TRIM25, but the beneficial roles of TRIM25 in modulating the immune response are still worth further exploration in duck.’ (Line 82-85).
Point 7: Lines 103, 206, 268, - Please italicize all scientific names (binomial nomenclature) of organisms
Response 7: Thank you for your suggestions. The scientific names of all organisms have been italicized.
Point 8: Line 137 – Reference should be provided for the pGL-IFN-β reporter plasmid
Response 8: Thank you for your suggestions. According to the sequence of duck IFN-β in NCBI gene pool (GenBank accession no. KM032183), part of the sequence of the duck IFN-β promoter region (- 167 to + 63) was selected to construct duIFN-β promoter reporter plasmid. The duIFN-β promoter was cloned into the KpnI and BglII sites of pGL3-Basic to form pGL3-IFN-β. We have added the method of constructing pGL-IFN-β reporter plasmid in Materials and Methods. (Line132-135).
Point 9: Line 143-145 – ELISA is mentioned in the methods- Results or Discussion do not mention the results of the Duck NF-κB and IRF1 ELISA. Please provide details.
Response 9: Sorry for the confusion and thanks for this helpful comment. To be honest, the ELISA method we described was indeed used in our subsequent experiments, but was not covered in the results of this article. It appeared in the article was a complete oversight on our part. We have deleted the content about ELISA. We apologize again for our mistake and thank you for reminding us.
Point 10: Line 263, 265, etc – ‘Flag’ – Please change to ‘FLAG’ while referring to the FLAG-tag
Response 10: Thank you for your suggestions. All ‘Flag’ which refer to the FLAG-tag have been changed to ‘FLAG’.

Round 2
Reviewer 1 Report
This reviewer thanks the authors for their point-by-point responses. While there are many ways to strengthen the manuscript and make better clarity for the readers (instead of relative to different tissue types, do a standard curve with known copy number), the manuscript can be presented after minor revision. In the authors response they reported that they can not do infection because of needing BSL3 facilities. Not all avian influenza viruses are BSL3 level, so this could be done in a BSL2. While the citations for 5’ triphosphate recognition are valid, a manuscript is always stronger with an actual infection if the authors are trying to relate their findings to a viral infection. This should be mentioned in the discussion that the findings in the context of an infection are yet to be determined.

Author Response
Point 1: This reviewer thanks the authors for their point-by-point responses. While there are many ways to strengthen the manuscript and make better clarity for the readers (instead of relative to different tissue types, do a standard curve with known copy number), the manuscript can be presented after minor revision. In the authors response they reported that they cannot do infection because of needing BSL3 facilities. Not all avian influenza viruses are BSL3 level, so this could be done in a BSL2. While the citations for 5’ triphosphate recognition are valid, a manuscript is always stronger with an actual infection if the authors are trying to relate their findings to a viral infection. This should be mentioned in the discussion that the findings in the context of an infection are yet to be determined.
Response 1: Thank you for your suggestions.
① To make better clarity for the readers, we changed the previous letter scheme. β-actin was used as the control gene.
Figure 3. (a) Relative expression levels of duTRIM25 in various tissues of five three-day-old healthy Shaoxing egg-laying ducks. The tissues tested included heart, liver, spleen, lung, kidney, duodenum, ovary, pancreas, muscle and blood. The mRNA expression levels were normalized to the expression of duck β-actin gene. Each bar represents the mean ± SE (n=5).
② Those comments are all valuable and very helpful for revising and improving our paper, as well as the important guiding significance to our further researches. In the next experiment, we will carry out some avian influenza infections that can be implemented in BSL2 to further verify our experimental results. In addition, we also made the following supplements in the discussion. ‘5’ppp dsRNA is a virus mimic, avian influenza virus infection can be simulated in vitro through it. It contains 3 phosphate groups on the 5'end as its key element, for the reason that can specifically activate the RLRs signaling pathway to induce type I interferon production. However, we must admit that there are still differences between the transfected virus mimics and avian influenza infection. Our research results provided a reference for the immune procedure in ducks after the simulation of avian influenza infection, but the findings in the actual infection situation have yet to be determined. In the next experiment, we will also use avian influenza infection to verify.’ (Line 347-350, page 10)

Reviewer 2 Report
The manuscript has been thoroughly revised by the authors incorporating the initial suggestions of my first round review.
1. The authors cited previous studies of duck TRIM25 and goose TRIM25 (Kaikai et al, 2021) & (Wei et al, 2016) and compared them with the current investigation.
2. The amino acid alignment and phylogenetic tree in figure 1 have been updated with the goose TRIM25 sequence.
3. Tissue expression of TRIM25 in healthy ducks has also been improved by the addition of blood tissue and using moderately TRIM25 expressing hearts as a control for comparison
4. Details of the gene expression primers are presented in the revised table 1
5. Further improvement to the manuscript was made by the updating of the Discussion and improvements to the language.
I recommend the revised version of the manuscript to be accepted. In order to make the manuscript even better, it would be great if the authors would correct the following minor errors listed below.
Minor errors to be corrected
Line 248 Salmonella Typhimurium, - change species name to 'typhimurium'
Line 332/ 337- renilla luciferase plasmids- ' change to 'Renilla" and italicize Renilla as its a genus name
Line 361 - Please correct Kai-Kai et al.[15] to Kaikai et al
Line 493 and 496- REFERENCES- Kindly update the references numbered 14 & 15 with correct author names (Last name) list . In the current version of the manuscript, I feel in the reference list the first name is listed here and it does not match with the inline mentions. References 14 & 15, should be listed as below with the correct last name as per the original citation for these papers. For #14 Yunan and #15 Han should be replaced by Wei and Kaikai respectively. Also check the names of other authors.
#14 . Wei Y, Zhou H, Wang A, Sun L, Wang M, Jia R, Zhu D, Liu M, Yang Q, Wu Y, Sun K, Chen X, Cheng A, Chen S. TRIM25 Identification in the Chinese Goose: Gene Structure, Tissue Expression Profiles, and Antiviral Immune Responses In Vivo and In Vitro. Biomed Res Int.
#15. Kaikai H, Zhao D, Liu Y, Liu Q, Huang X, Yang J, Zhang L and Li Y (2021) The E3 Ubiquitin Ligase TRIM25 Inhibits Tembusu Virus Replication in vitro. Front. Vet. Sci. 8:722113.
Author Response
Point 1: The manuscript has been thoroughly revised by the authors incorporating the initial suggestions of my first round review.
- The authors cited previous studies of duck TRIM25 and goose TRIM25 (Kaikai et al, 2021) & (Wei et al, 2016) and compared them with the current investigation.
- The amino acid alignment and phylogenetic tree in figure 1 have been updated with the goose TRIM25 sequence.
- Tissue expression of TRIM25 in healthy ducks has also been improved by the addition of blood tissue and using moderately TRIM25 expressing hearts as a control for comparison
- Details of the gene expression primers are presented in the revised table 1
- Further improvement to the manuscript was made by the updating of the Discussion and improvements to the language.
I recommend the revised version of the manuscript to be accepted. In order to make the manuscript even better, it would be great if the authors would correct the following minor errors listed below.
Response 1: Thank you for your suggestions. Those comments are all valuable and very helpful for revising and improving our paper, as well as the important guiding significance to our further researches. We have carefully revised the manuscript according to the comments and provided point-to-point responses to all the questions raised by the reviewers below.
Point 2: Line 248 Salmonella Typhimurium, - change species name to ' typhimurium '
Response 2: Thank you for your suggestions. We have changed Salmonella Typhimurium to Salmonella typhimurium.
Point 3: Line 332/ 337- renilla luciferase plasmids- ' change to 'Renilla" and italicize Renilla as its a genus name
Response 3: Thank you for your suggestions. We have changed renilla luciferase plasmids to Renilla luciferase plasmids and italicized it as a genus name.
Point 4: Please correct Kai-Kai et al. [15] to Kaikai et al
Response 4: Thank you for your suggestions. We have corrected Kai-Kai et al. [15] to Kaikai et al.
Point 5: Line 493 and 496- REFERENCES- Kindly update the references numbered 14 & 15 with correct author names (Last name) list. In the current version of the manuscript, I feel in the reference list the first name is listed here and it does not match with the inline mentions. References 14 & 15, should be listed as below with the correct last name as per the original citation for these papers. For #14 Yunan and #15 Han should be replaced by Wei and Kaikai respectively. Also check the names of other authors.
#14. Wei Y, Zhou H, Wang A, Sun L, Wang M, Jia R, Zhu D, Liu M, Yang Q, Wu Y, Sun K, Chen X, Cheng A, Chen S. TRIM25 Identification in the Chinese Goose: Gene Structure, Tissue Expression Profiles, and Antiviral Immune Responses In Vivo and In Vitro. Biomed Res Int.
#15. Kaikai H, Zhao D, Liu Y, Liu Q, Huang X, Yang J, Zhang L and Li Y (2021) The E3 Ubiquitin Ligase TRIM25 Inhibits Tembusu Virus Replication in vitro. Front. Vet. Sci. 8:722113.
Response 5: Thank you for your suggestions. We checked the reference format of GENES and the latest articles, and found that the author's last name was written first, and the first name was written last. In order to match the contents of the manuscript with the references, we have replaced Kaikai et al. in the paper with Han et al. In addition, we have carefully checked the author names of all cited articles to ensure that they match the references. If our understanding was wrong, please further criticize and correct.
#14. Wei, Y.N.; Zhou, H.; Wang, A.; Sun, L.P.; Wang, M.S.; Jia, R.Y.; Zhu, D.K.; Liu, M.F.; Yang, Q.; Wu, Y.; et al. TRIM25 Identification in the Chinese Goose: Gene Structure, Tissue Expression Profiles, and Antiviral Immune Responses In Vivo and In Vitro. BioMed research international 2016, 2016, 1403984.
#15. Han, K.K.; Zhao, D.; Liu, Y.; Liu, Q.; Huang, X.; Yang, J.; Zhang, L.; Li, Y. The E3 Ubiquitin Ligase TRIM25 Inhibits Tembusu Virus Replication in vitro. Front Vet Sci 2021, 8, 722113.
